Candidate pathogenicity islands in the genome of ‘Candidatus Rickettsiella isopodorum’, an intracellular bacterium infecting terrestrial isopod crustaceans

Wang YaDong 1 2
Chandler Christopher christopher.chandler@oswego.edu 1
1 Department of Biological Sciences, State University of New York at Oswego , Oswego , NY , United States
2 Department of Biological Sciences, State University of New York at Buffalo , Buffalo , NY , United States
Chrostek Ewa
Electronic publication date: 2016 Dec 21
Publication date: 2016
Volume: 4
Electronic Location ID: e2806
Received 2016 Aug 5; Accepted 2016 Nov 20
Copyright: ©2016 Wang and Chandler
Copyright year: 2016
Copyright holder: Wang and Chandler
License: This is an open access article distributed under the terms of the Creative Commons Attribution License, which permits unrestricted use, distribution, reproduction and adaptation in any medium and for any purpose provided that it is properly attributed. For attribution, the original author(s), title, publication source (PeerJ) and either DOI or URL of the article must be cited.
License URL: https://creativecommons.org/licenses/by/4.0/

Keywords: Rickettsiella, Genomic islands, Trachelipus rathkei, mcf2

Funding: Rice Creek Associates SUNY Oswego Scholarly & Creative Activities Committee NSF DEB-1453298 This work was supported by a Rice Creek Associates Small Grant to CHC, a SUNY Oswego Scholarly and Creative Activities Committee grant to YW, and NSF DEB-1453298 to CHC. The funders had no role in study design, data collection and analysis, decision to publish, or preparation of the manuscript.

==============================
The bacterial genus Rickettsiellabelongs to the order Legionellales in the Gammaproteobacteria, and consists of several described species and pathotypes, most of which are considered to be intracellular pathogens infecting arthropods. Two members of this genus, R. grylliand R. isopodorum, are known to infect terrestrial isopod crustaceans. In this study, we assembled a draft genomic sequence for R. isopodorum, and performed a comparative genomic analysis with R. grylli. We found evidence for several candidate genomic island regions in R. isopodorum, none of which appear in the previously available R. grylli genome sequence.Furthermore, one of these genomic island candidates in R. isopodorum contained a gene that encodes a cytotoxin partially homologous to those found in Photorhabdus luminescensand Xenorhabdus nematophilus (Enterobacteriaceae), suggesting that horizontal gene transfer may have played a role in the evolution of pathogenicity in Rickettsiella. These results lay the groundwork for future studies on the mechanisms underlying pathogenesis in R. isopodorum, and this system may provide a good model for studying the evolution of host-microbe interactions in nature.

Introduction

Rickettsiella is a genus of bacteria that infects a range of arthropod hosts, including insects, crustaceans, and arachnids (Dutky & Gooden, 1952; Hall & Badgley, 1957; Vago & Martoja, 1963; Vago et al., 1970; Leclerque & Kleespies, 2008a; Leclerque & Kleespies, 2012; Tsuchida et al., 2010; Kleespies et al., 2011; Leclerque et al., 2011a). Originally, this genus of bacteria was classified as a member of the order Rickettsiales in the Alphaproteobacteria based on ultrastructural analyses (Hall & Badgley, 1957; Vago & Martoja, 1963; Vago et al., 1970). However, based on 16S rRNA gene sequence analyses, this genus has been recently reclassified to the order Legionellales in the Gammaproteobacteria, which also includes human pathogens such as Coxiella and Legionella (Roux et al., 1997; Cordaux et al., 2007; Leclerque, 2008; Leclerque & Kleespies, 2008b; Leclerque & Kleespies, 2008c). There are several described species in the genus Rickettsiella: R. popilliae (Dutky & Gooden, 1952), R. grylli (Vago & Martoja, 1963), R. chironomi (Weiser & Žižka, 1968), R. stethorae (Hall & Badgley, 1957), plus the recently described ‘Candidatus Rickettsiella isopodorum’ (Kleespies, Federici & Leclerque, 2014) and ‘Candidatus Rickettsiella viridis’ (Tsuchida et al., 2014). There are also several additional pathotypes thought to be synonyms of the previously recognized species, including R. tipulae (Leclerque & Kleespies, 2008a), R. agriotidis (Leclerque et al., 2011a), R. pyronotae (Kleespies et al., 2011), and R. costelytrae (Leclerque et al., 2012). In addition, another bacterium designated as Diplorickettsia, found in ticks, is also closely related and may actually belong in this genus (Mediannikov et al., 2010; Iasur-Kruh et al., 2013). There is probably much greater diversity in this genus than currently recognized; for instance, one genetic study identified multiple distinct lineages of Rickettsiella in just one species of tick, Ixodes uriae (Duron, Cremaschi & McCoy, 2015).

Although the full breadth of Rickettsiella diversity has yet to be discovered, the most thoroughly characterized species of this genus are believed to be intracellular arthropod pathogens (Dutky & Gooden, 1952; Hall & Badgley, 1957; Vago et al., 1970; Abd El-Aal & Holdich, 1987). In terrestrial isopods, symptoms of infection include white viscous fluid filling the body cavity of the host, weight loss, and death (Vago et al., 1970). However, there are cases in which Rickettsiella has more benign effects. For example, in pea aphids (Acyrthosiphon pisum), the presence of Rickettsiella spp. is correlated with the intensity of green coloration, potentially helping evade predators (Tsuchida et al., 2010). In another study on pea aphids, Rickettsiella was shown to have neutral or positive effects on host fitness, depending on host genotype and whether the host was coinfected with Hamiltonella (Tsuchida et al., 2014).

A recent study characterized Rickettsiella-like infections in terrestrial isopods and described a new species, ‘Candidatus Rickettsiella isopodorum’ (Kleespies, Federici & Leclerque, 2014). This isopod pathogen seems to have two distinct cell types—larger but variably sized cells representing an earlier developmental phase, and smaller, rod-shaped cells representing the infectious form—similar to other Rickettsiella infections. Unlike insect Rickettsiella infections, however, it lacks well-defined membrane-bound protein crystals. This study also used multiple genetic markers to explore strain variation between Rickettsiella samples isolated from two host species, Armadillidium vulgare and Porcellio scaber, from two geographically distinct locations: California, USA and Germany, respectively. Based on genetic data, ‘Candidatus Rickettsiella isopodorum’ was clearly nested within the genus Rickettsiella, and the sequences obtained from the two host samples were nearly identical, even though they were from different host species isolated from localities separated by thousands of miles.

Outside of these phylogenetic studies, however, few genetic data from Rickettsiella are available. There is one publicly available genome sequence for a R. grylli isolate from a terrestrial isopod (GenBank Accession Number NZ_AAQJ00000000); however, the source of this specimen is annotated only as “pillbugs from back yard”, so the exact host species it was derived from, and what symptoms it displayed, are unclear. Moreover, this R. grylli sequence clearly belongs to a distinct lineage from ‘Candidatus Rickettsiella isopodorum’ (Kleespies, Federici & Leclerque, 2014). Although this genome has been used in phylogenomic studies (Leclerque, 2008) and does possess a full icm/dot type IV secretion system (Zusman et al., 2007), its gene content nevertheless remains mostly unexplored, and comparative genomic analyses have been impossible in the absence of whole-genome sequences from other Rickettsiella lineages.

The aim of this study was to explore genomic variation among Rickettsiella isolates in terrestrial isopods. We assembled a draft genome sequence of Rickettsiella isopodorum using high-throughput sequencing data obtained from a wild-caught terrestrial isopod. We then performed comparative genomic analyses between R. isopodorum and R. grylli. Specifically, we looked for evidence of genomic islands. We identified several such candidate genomic regions that appear to be present in R. isopodorum but not in R. grylli or another outgroup lineage, Diplorickettsia massiliensis. Some of these regions appear to contain genes with possible roles in pathogenicity, such as cytotoxins. These findings lay the groundwork for future studies of genetic and phenotypic diversity in this widespread arthropod endosymbiont.

Methods & Results

Genome assembly and annotation

One male and one female specimen of Trachelipus rathkei were wild-caught at Rice Creek Field Station in Oswego, NY, USA. Genomic DNA was isolated from head, leg, gonad, and ventral tissue from each specimen using a Qiagen DNeasy Blood & Tissue Kit (Qiagen) and submitted to the University at Buffalo Next-Generation Sequencing Core Facility. The samples were barcoded and sequenced in a single 2 × 100 rapid-run lane on an Illumina HiSeq 2500, with the initial goal of obtaining pilot data for another project. Raw sequence reads were deposited into the NCBI Short Read Archive (SRA) under accession numbers SRR4000573 and SRR4000567.

Raw sequence reads were filtered for quality and sequencing adapters were removed using Trimmomatic v. 0.35 (Bolger, Lohse & Usadel, 2014), with a minimum leading and trailing sequence quality of 5, and a minimum internal sequence quality of 5 using a sliding window of 4 bp. An initial de novo assembly was performed with Minia 1.6906 (Chikhi & Rizk, 2012; Salikhov, Sacomoto & Kucherov, 2014), with a k-mer size of 53 and a minimum k-mer depth of 4.

Inspection of the resulting contigs revealed the presence of sequences from two distinct Rickettsiella-like endosymbionts. One set of contigs displayed high (>95%) sequence similarity to the Rickettsiella grylli sequence in Genbank (NZ_AAQJ00000000) and were more abundant in the male sample (∼20× coverage in the male, <1× coverage in the female; Fig. 1, red dots). The other set showed lower sequence similarity to R. grylli NZ_AAQJ00000000 (∼70–80%) and were more abundant in the female (>1,000× coverage vs. <5× coverage in the male; Fig. 1, blue dots). These sets of contigs summed to total lengths of ∼1.3 Mb and ∼1.4 Mb, respectively, just slightly shorter than the ∼1.6 Mb R. grylli NZ_AAQJ00000000 sequence, consistent with the hypothesis that this initial assembly contained nearly complete genome sequences from two distinct Rickettsiella strains.

Figure 1 Rickettsiella contigs in initial assembly.

Initial assembly of all Trachelipus rathkei sequencing data seems to contain contigs coming from two distinct Rickettsiella lineages. (A) One set of contigs displays high similarity to R. grylli NZ_AAQJ00000000 (red dots), while the contigs in the other set are more divergent (blue dots), but contigs in both sets seem to span the entire length of the R. grylli genome. (B) The two sets of contigs also form two distinct clusters based on sequencing depth in the two T. rathkei samples; the high similarity contigs are present at moderate depth in the male sample and very low depth in the female sample, while the low similarity contigs are present at very high depth in the female sample and low depth in the male sample; only a small number of contigs seem to be mis-classified.

We suspected that co-assembling these genomes together might result in “tangled” deBruijn graphs due to their similarity, resulting in fragmented assemblies in spite of moderate to high coverage, especially for the more divergent strain with higher coverage. To ameliorate this problem, we developed a custom pipeline for an iterative mapping-and-assembly strategy, somewhat similar to approaches used by others to assemble mitochondrial genomes from whole genomic DNA (Hahn, Bachmann & Chevreux, 2013; Rivarola-Duarte et al., 2014; Chandler et al., 2015). We first divided the contigs into two sets, representing each strain of Rickettsiella, based on sequence coverage in the two DNA samples, as partitioning them based on homology to R. grylli NZ_AAQJ00000000 alone seemed to mis-classify a few contigs (Fig. 1B). We then mapped the sequence reads to these contigs, identifying the subset of sequence reads (and their paired reads) derived from each of these two putative Rickettsiella strains. For the first set, representing the strain similar to R. grylli NZ_AAQJ00000000, we kept only the reads from the male isopod sample, as this was the dominant Rickettsiella strain in this sample; likewise, for the second set, representing the more divergent Rickettsiella strain, we kept only the reads from the female isopod, in which this strain was more abundant. These two sets of sequence reads were then assembled separately to generate two new assemblies, using SOAPdenovo2 (Luo et al., 2012) with a k-mer size of 53. Because these second assemblies might still contain some gaps, we repeated this process 10 times, mapping all sequence reads to the assembly and keeping any reads that mapped along with their “mates” after each iteration. After the last iteration, we took the sets of sequence reads from each putative Rickettsiella strain and re-assembled each set using a variety of different values for the parameters k (k-mer size; 41, 45, 53, 59, 61, 63, and 67) and d (minimum k-mer depth; 5, 10, and 20 for the higher-coverage, more divergent Rickettsiella; and 2, 3, and 4 for the lower-coverage, more similar Rickettsiella). Finally, we evaluated all the candidate final assemblies using QUAST v.2.2 (Gurevich et al., 2013). From each set of assemblies, we selected the assembly with the largest N50 as the optimal final assembly (Table 1); in each case, the assembly with the largest N50 was also the longest or only slightly shorter than the longest.

Table 1 Assembly statistics for Rickettsiella grylli and Rickettsiella isopodorum genomes assembled from Trachelipus rathkei sequencing data.

Pseudogenes includes incomplete or partial gene sequences, which may include functional genes that are falsely predicted to be pseudogenes because they are fragmented or only partially assembled; the number in parentheses indicates the number of predicted pseudogenes with frameshift mutations or internal stop codons, excluding partially assembled genes.

	R. grylli	R. isopodorum	
Number of contigs	851	198	
Total length (bp)	1,369,903	1,509,158	
N50 (bp)	5,907	384,641	
Longest contig (bp)	33,090	583,532	
GC content (%)	38.32	37.06	
Total genes	1,454	1,330	
Pseudogenes	337 (9)	39 (3)	
rRNAs	1 complete 5S, 2 partial 16S, 2 partial 23S	1 complete 16S, 1 complete 23S	
tRNAs	34	40	
ncRNAs	4	4	

We evaluated our assemblies using REAPR v.1.0.18 (Hunt et al., 2013) to check for errors. REAPR looks for various types of anomalies in paired sequence reads mapped back to the assembly, and splits scaffolds or inserts gaps into contigs where such anomalies occur, as they are possible indicators of misassembled regions. REAPR identified several possible errors in each assembly; these potential errors did not span gaps in the assembly, indicating that they are likely insertions or deletions rather than mis-joins (REAPR manual). To further investigate the nature of these errors, we extracted the sequences of a random subset of the candidate errors in our R. grylli assembly, along with their flanking regions, and aligned them to the R. grylli NZ_AAQJ00000000 assembly using BLAST+ v.2.3.0 (Camacho et al., 2009). In all cases, they aligned perfectly (Fig. S1), suggesting that these regions are correctly assembled in spite of being flagged as errors. Moreover, we visually inspected mapped read pairs in predicted error regions using IGV (Robinson et al., 2011; Thorvaldsdóttir, Robinson & Mesirov, 2013), and in all cases, reads appeared to map to these regions normally, but with lower coverage; there was no evidence of discordantly mapped read pairs (e.g., in the wrong orientation or paired reads mapping to different contigs) (Fig. S2). We obtained similar results when manually inspecting regions flagged as errors by REAPR in our R. isopodorum assembly (Figs. S3 and S4). We therefore concluded that most errors reported by REAPR in these assemblies may simply be false positives resulting from lower-than-typical sequencing depth in those regions, which could have various causes, such as sequencing bias due to variation in GC content (Ekblom, Smeds & Ellegren, 2014). Therefore, we performed subsequent analyses using the original assemblies, not the assemblies generated by REAPR in which possible errors are replaced by gaps.

Despite our efforts to improve the assemblies, the final versions still remained somewhat fragmented. In particular, although most of the sequence data in the R. isopodorum assembly was found in a few very large contigs, the total assembly still contained numerous short contigs. These might consist of sequences found in multiple copies in the R. isopodorum genome, or they might perhaps be sequence fragments from the host isopod or other other bacteria present in the DNA sample. To explore this further, we created taxon-annotated GC-coverage plots (blobplots) using blobtools v0.9.19 (Kumar et al., 2013). A blobplot of an assembly of all of the sequence data from the female T. rathkei isopod sample showed a single high-coverage cluster of contigs matching proteobacteria, representing the R. isopodorum genome (Fig. S5). Though there were a few other “blobs” of bacteria evident in this graph, including at least one other genome from the proteobacteria and one from actinobacteria, these are not likely to be misassembled with the R. isopodorum genome because of their much lower coverage and higher GC content. We also examined a blobplot created from the final R. isopodorum assembly (Fig. S6). This plot showed one relatively short contig with higher coverage and GC content than the rest of our assembly; however, a BLAST search using this sequence against the nt database revealed that it closely matches 16S and 23S rRNA sequences from other Rickettsiella isolates, so it is unlikely to represent a contaminant sequence. Instead, rRNA operons are frequently found in multiple copies in bacterial genomes, explaining the higher coverage of this contig as well as why it did not assemble into a longer contig. The remaining short contigs, totaling just 0.02 Mb, also had higher coverage than the rest of our genome, suggesting that these are also multi-copy sequences; indeed, several of these had close BLAST hits to multiple locations in the R. grylli genome (Fig. S7). Therefore, the fragmentation of our final assembly can probably be explained by the presence of repeat sequences. Moreover, this assembly appears to contain few, if any, contaminating sequences from other bacteria.

The draft genome sequences were deposited in GenBank under accession numbers MCRF00000000 and LUKY00000000 and annotated with the NCBI Prokaryotic Genome Annotation Pipeline (Angiuoli et al., 2008). We also used ISsaga (Varani et al., 2011) and PHASTER (Arndt et al., 2016) to look for potential insertion sequences and prophage elements; two candidate partial insertion sequences (Fig. S8) and one candidate prophage element (Fig. S9) were found. Finally, we conducted a preliminary pathway anaysis (Supplementary Data) using BlastKOALA (Kanehisa, Sato & Morishima, 2016).

Phylogenetic analyses

To examine the phylogenetic relationships between our two Rickettsiella strains and those identified in other studies, we downloaded from GenBank the sequences of four genes (ftsY, gidA, rpsA, and sucB) from ‘Candidatus Rickettsiella isopodorum’ (accession numbers JX406181, JX406182, JX406183, JX406184) previously used for multi-locus sequence typing (Kleespies, Federici & Leclerque, 2014). We used these sequences as queries in tblastx searches in our two assembled genomes. Every query produced a clear single best hit in each genome, so we extracted the sequences of these four genes from our assembled genomes to use in phylogenetic analyses. We also extracted the same gene sequences from the previously sequenced genomes of Rickettsiella grylli (NZ_AAQJ00000000) and Diplorickettsia massiliensis (NZ_AJGC00000000). In addition, we downloaded sequences from several other Rickettsiella isolates from a variety of host species. For each gene, we aligned the sequences using ClustalOmega (Sievers et al., 2011) with the default parameters, concatenated the aligned sequences from each gene, and then used the GBlocks server (Castresana, 2000; Talavera & Castresana, 2007) to remove poorly aligned regions. The final alignment contained a total of 3290 bases. Finally, we constructed a phylogenetic tree in MEGA7 (Kumar, Stecher & Tamura, 2016) using maximum likelihood with the Tamura-Nei model (Tamura & Nei, 1993), which is the default option. Node support was estimated by bootstrapping with 100 replicates.

Our first Rickettsiella genome, found at moderate coverage in the male T. rathkei sample, clustered closely with the R. grylli NZ_AAQJ00000000 genome sequence from GenBank with 100% bootstrap support (Fig. 2A), consistent with the high sequence similarity we initially observed between scaffolds from this assembly and the R. grylli genome. Our second Rickettsiella genome, found at very high sequencing depth in the female T. rathkei specimen, on the other hand, clustered closely with ‘Candidatus Rickettsiella isopodorum’ also with 100% bootstrap support (Fig. 2A), suggesting that this represents the genome of ‘Candidatus Rickettsiella isopodorum’.

Figure 2 Phylogenies.

(A) Phylogeny based on ftsY, gidA, rpsA, and sucB sequences from the two Rickettsiella genomes assembled from Trachelipus rathkei, R. grylli NZ_AAQJ00000000, D. massiliensis, and other Rickettsiella sequences from other phylogenetic studies. Phylogenies were generated in MEGA7 using maximum likelihood with the Tamura-Nei model, and node support was estimated using bootstrapping with 100 replicates. (B) Phylogeny based on gidA sequences from the same samples as (A), with the addition of several other isopod samples from upstate New York in which gidA sequences were obtained by PCR and Sanger sequencing.

To further explore Rickettsiella diversity from the upstate New York region, we captured additional isopods from Oswego, NY and around Syracuse, NY. DNA was isolated from tissues following the same protocols as above. PCRs were performed using primers to amplify a portion of the gidA gene, using the same primer sequences and PCR conditions as in previously published studies (Leclerque et al., 2011b; Kleespies, Federici & Leclerque, 2014). PCR products from positive samples were cleaned using ExoSAP-IT (Affymetrix) and submitted to Genewiz (South Plainfield, NJ) for Sanger dye-terminator sequencing. These sequences were manually cleaned using 4Peaks (Nucleobytes) and aligned with the gidA sequences from the dataset described above. Poorly aligned regions were again removed using GBlocks, and a phylogeny was constructed using the same procedure as earlier.

Most of the upstate NY samples clustered closely with the ‘Candidatus Rickettsiella isopodorum’ sequence that we assembled (Fig. 2B), including samples from a variety of other isopod species, including Oniscus asellus, Armadillidium vulgare, Porcellio scaber, and Philoscia muscorum. However, amplified gidA sequences from two samples (one T. rathkei and one P. muscorum) clustered closely with our assembled Rickettsiella grylli sequence (Fig. 2B).

Synteny and genomic islands

Given that our assembled R. grylli sequence was very fragmented (>800 contigs, N50 5.9 kb; Table 1) and highly similar to the previously available R. grylli sequence, we instead chose to focus our comparative analyses on the more divergent and contiguous R. isopodorum assembly that we obtained. Genomic islands are generally identified using two basic strategies: (i) analyses of sequence composition, looking for regions that have unusual nucleotide frequencies compared to the rest of the genome; and (ii) comparative approaches (Langille, Hsiao & Brinkman, 2010). Our initial attempts to use sequence composition-based approaches were unsuccessful, with high false positive rates (e.g., some software tools classified nearly half of the genome as “islands”). Moreover, most comparative tools are designed to be used with more complete assemblies (e.g., scaffolds representing whole chromosomes) and/or require a collection of closely related genome sequences for comparison. Because we only had two reference genomes for comparison (R. grylli NZ_AAQJ00000000 and D. massiliensis NZ_AJGC00000000; Mathew et al., 2012), and because our R. isopodorum assembly as well as one of our reference genomes were still somewhat fragmented (divided into 6 or more scaffolds), we were precluded from using most “off-the-shelf” tools. Therefore, we developed a custom pipeline aimed at finding candidate genomic islands in our R. isopodorum assembly, by screening for regions in our assembly that apparently lack homologs in both R. grylli and D. massiliensis.

We first broke our assembled R. isopodorum sequences into 200-bp chunks. Each of these chunks was used as a query in blastn and tblastx searches against the R. grylli genome, using an e-value threshold of 1 × 10−6. We examined patterns of synteny between the two genomes by creating dot plots, plotting the position of each query sequence from our R. isopodorum assembly against the location of its match or matches in R. grylli. Next, we searched for consecutive sets of “chunks” of the R. isopodorum genome that had no blastn or tblastx hits in the R. grylli genome at this threshold, to identify candidate genomic islands in R. isopodorum. This process was repeated against the D. massiliensis genome. We focused our subsequent analyses on the strongest candidate genomic islands, identified as those regions in R. isopodorum that had no apparent homology to any portion of either the R. grylli or D. massiliensis genomes.

We observed extensive synteny between our R. isopodorum assembly and R. grylli (Fig. 3A), but less so between R. isopodorum and D. massiliensis (Fig. 3B), which is not surprising given the more distant relationship in this latter case. We found 8 candidate genomic islands (Table 2) ranging in size from ∼3.6 kb to ∼11 kb. We used the nucleotide sequences of these entire regions, and the sequences of predicted proteins within these regions, as queries in blastn and tblastx searches against the R. grylli and D. massiliensis genomes. None of these subsequent searches turned up close hits except for two proteins that did have weak matches. In some cases, the prediction that these regions are genomic islands was further supported by GC content drastically different from the rest of the genome (27–28% vs. 37%). In several cases, predicted gene/protein sequences had no matches in available nucleotide or protein databases. In other cases, predicted genes showed homology to genes annotated in other bacterial genomes but not in R. grylli or D. massiliensis.

Figure 3 Synteny and genomic islands.

Dot plots showing synteny between Rickettsiella isopodorum and (A) R. grylli and (B) Diplorickettsia massiliensis. Light gray lines indicate borders between contigs in each assembly. Vertical pink bars indicate candidate genomic island regions, i.e., sequences in R. isopodorum that have no matches in blastn or tblastx searches against each reference genome.

Table 2 Candidate genomic island regions in R. isopodorum.

% Cov.: percentage of query sequence that aligns to the best BLAST hit; % Id.: percentage of aligned amino acids that are identical to best BLAST hit; % Pos.: percentage of positive-scoring amino acids in aligned region of best BLAST hit.

Predicted location	Size (bp)	GC (%)	Predicted genes & functional notes	Size (aa)	% Cov.	% Id.	% Pos.	
contig_191: 18,201–21,800	3,600	27.3	A1D18_00540: no apparent homology	209	n/a	n/a	n/a	
A1D18_00545: no apparent homology	105	n/a	n/a	n/a	
A1D18_00550: no apparent homology	61	n/a	n/a	n/a	
A1D18_00555: no apparent homology	103	n/a	n/a	n/a	
A1D18_00560: portion matches hypothetical protein in Diplorickettsia	274	50	35	53	
contig_193: 31,601–36,400	4,800	28.0	A1D18_00930: no apparent homology	329	n/a	n/a	n/a	
A1D18_00935: no apparent homology	428	n/a	n/a	n/a	
A1D18_00940: no apparent homology	448	n/a	n/a	n/a	
A1D18_00945: no apparent homology	149	n/a	n/a	n/a	
contig_196: 42,801–48,200	5,400	31.0	A1D18_01965: matches hypothetical protein from nucleopolyhedrovirus virus from Lepidoptera (e = 7e−27); also matches hypothetical protein and surface-related protein entries from Ehrlichia (e = 7e−04); contains a predicted peptidase M9 domain, which is predicted to break down collagens	1,213	40	25	41	
contig_196: 85,801–89,600	3,800	37.9	A1D18_02135: has matches in Legionella, Pseudomonas, Hahella, Streptomyces; contains predicted polyketide synthase domain	1,862	98	40	58	
contig_197: 50,601–54,600	4,000	33.5	A1D18_02420: matches permease from Yersinia (e = 0)	452	99	75	85	
A1D18_02425: matches a hypothetical protein in Rickettsiella grylli	412	85	26	44	
A1D18_02430: no apparent homology	255	n/a	n/a	n/a	
contig_197: 72,601–83,600	11,000	35.5	A1D18_02500: part of gene is outside of predicted island; has partial match to a hypothetical protein in Diplorickettsia massiliensis (44% coverage), but better matches to proteins in Pseudomonas (63% coverage); matches are annotated as adenylate cyclase and anthrax toxin; contains predicted Anthrax toxin domain	545	62	38	51	
A1D18_02505: Contains a predicted RING domain, a type of zinc finger domain implicated in many functions, and a Ubox domain, implicated in ubiquitination; matches are in eukaryotes, not prokaryotes	313	18	34	50	
A1D18_02510: Matches Mcf2, cytotoxin from Xenorhabdus nematophila and Photorhabdus luminescens; contains TcdA_TcdB pore domain; also matches Clostridium difficile toxin	2,928	33	27	47	
contig_197: 278,601–285,600	7,000	33.6	A1D18_03425; has three domains common to thyamine pyrophosphate enzymes; top hit is an uncultured bacterium, but secondary matches in Pseudomonas	628	95	69	80	
A1D18_03430; contains predicted phosphate binding domain, aldolase domain; matches uncultured bacterium, Flavobacterium, Pedobacter, Janthinobacterium, Oxalobacteraceae, Paenibacillus, Planktothrix	335	99	73	88	
A1D18_03435; matches predicted acetaldehyde dehydrogenase enzymes from same taxa as A1D18_03430	294	96	67	80	
A1D18_03440; matches predicted dolichol phosphate mannose synthase enzymes from Legionella, Paenibacillus, Tatlockia, and other bacteria	311	99	61	78	
A1D18_03445: matches UDP-glucuronate decarboxylase enzymes from Pseudanabaena, Planktothrix	355	97	60	76	
A1D18_03450: matches polysaccharide biosynthetase (synthesizes cell surface polysaccharides) from Paenibacillus, Legionella, Sulfuricurvum	147	76	46	67	
A1D18_03455; matches pyridoxal phosphate (PLP)-dependent aspartate aminotransferase superfamily proteins from Tatlockia, Legionella	400	100	71	84	
contig_198: 245,601–250,400	4,800	37.3	None predicted by annotation software; however, tblastn searches of this sequence show two portions matching glycosyltransferase enzymes from Herbasperillum, Serratia, and Ochtrobactrum	n/a	n/a	n/a	n/a	

One genomic island, in particular, contained a gene showing homology to the makes caterpillars floppy (Mcf) family of toxin genes from Photorhabdus luminescens and Xenorhabdus nematophilus (Daborn et al., 2002; Waterfield et al., 2003), which has also been horizontally transferred into some fungal symbionts of plants (Ambrose, Koppenhöfer & Belanger, 2014); this gene had no hits in any members of the Legionellales. To look for potential donors for this putatively horizontally transferred gene in R. isopodorum, we downloaded several Mcf and related amino acid sequences from Genbank, and aligned them and constructed a phylogeny using the same methods as described above, except using the JTT matrix-based model because this dataset consisted of amino acid sequences, not nucleotides (Jones, Taylor & Thornton, 1992). The R. isopodorum homolog of this gene did not align as well to these sequences as they did to each other; after filtering out poorly aligned sites using GBlocks (Castresana, 2000; Talavera & Castresana, 2007), only 1,190 aligned amino acids (out of 2,928 amino acids in the predicted R. isopodorum sequence) remained, including the conserved TcdA/TcdB pore-forming domain. Consistent with this, the R. isopodorum Mcf-like gene did not cluster closely with any of the other members of this family; instead, it fell in its own clade, separated from the others by a long branch (Fig. 4).

Figure 4 Mcf phylogeny.

Phylogenetic tree showing inferred relationships among Mcf-like genes, obtained via maximum likelihood using the JTT matrix-based model. Numbers indicate bootstrap support using 100 replicates. No outgroup was specified in this analysis; instead, the tree was rooted at the longest branch.

To further test for horizontally transferred genes, we also used HGTector v.0.2.1 (Zhu, Kosoy & Dittmar, 2014). In this analysis, we used the NCBI non-redundant protein database (nr) as the ‘distal’ group. Initial attempts to include R. grylli, D. massiliensis, and other Legionella and Coxiella genomes as the ‘close’ group resulted in a large number of candidate horizontally transferred genes (>10% of the predicted genes), many of which actually had BLAST hits in other members of the Legionellales, so we performed a more restrictive analysis using only R. grylli and D. massiliensis as the ‘close’ group, which yielded only 11 candidate horizontally transferred genes (Table 3). Only one of these candidate genes was also identified in our genomic island analysis. However, this discrepancy is likely due to the slightly different aims of these two approaches: our genomic island analysis was designed to identify any large novel genomic regions in R. isopodorum by looking for sequence regions that are absent in close relatives, regardless of similarity to potential donors; HGTector, on the other hand, requires close relatives of the donor species to be present in the reference database. Indeed, many of the predicted candidate genes in the candidate genomic islands we identified initially had no good blast hits in the nr database (Table 2), explaining why they were not identified by HGTector. There are also several possible explanations for why HGTector detected several candidates not identified in our custom analysis. For example, our analysis focused on regions of R. isopodorum of at least 3,000 bp in size without matches in either R. grylli or D. massiliensis; therefore, it is not surprising that it missed the smaller, mostly single-gene putative transfer events identified by HGTector. In addition, HGTector is based on protein BLAST searches of predicted amino acid sequences, not the genome sequence itself, whereas our custom approach also includes nucleotide BLAST searches of close relatives. Therefore, if the annotations of R. grylli and D. massiliensis are missing some genes, these would appear to be absent from these genomes to HGTector, and might therefore turn up as false candidates for horizontally transferred genes.

Table 3 Candidate horizontally transferred genes identified by HGTector.

R. isopodorum protein	Closest match	Functional notes	Size (aa)	% Cov.	% Id.	% Pos.	
A1D18_00810	Rhizobium leucaenae (Rhizobiales, Alphaproteobacteria)	Outer membrane autotransporter; contains an autotransporter and a pertactin-like passenger domain; proteins in this family are usually virulence factors	932	64	43	60	
A1D18_02160	Legionella shakespearei (Legionellales, Gammaproteobacteria)	NAD/FAD binding protein	236	92	53	70	
A1D18_03435	Paenibacillus algorifonticola (Bacillales, Bacilli)	acetaldehyde dehydrogenase; also identified in genomic island analysis	294	96	67	80	
A1D18_03465	Beggiatoa alba (Thiotrichales, Gammaproteobacteria)	glucose-1-phosphate cytidylyltransferase	268	95	71	85	
A1D18_03485	Escherichia coli (Enterobacteriales, Gammaproteobacteria)	rhamnosyltransferase	313	95	30	49	
A1D18_03490	Acinetobacter sp. NCu2D-2 (Pseudomonodales, Gammaproteobacteria)	rhamnosyltransferase	289	93	30	52	
A1D18_03505	Sulfuritalea hydrogenivorans sk43H (Rhodocyclales, Betaproteobacteria)	glycosyltransferase	268	98	36	58	
A1D18_05025	Arenimonas composti (Xanthomonodales, Gammaproteobacteria)	glycosyltransferase	409	96	38	63	
A1D18_05065	Aeromonas tecta (Aeromonodales, Gammaproteobacteria)	N-acetyltransferase	146	93	36	57	
A1D18_06515	Shewanella pealeana (Alteromonadales, Gammaproteobacteria)	aquaporin	230	99	69	79	
A1D18_06535	Hahella chejuensis (Oceanospirillales, Gammaproteobacteria)	matches hypothetical proteins; contains Permuted papain-like amidase enzyme	256	93	53	76	

Discussion

Assembly of bacterial genomes from mixed species data

Our results demonstrate that it is feasible to assemble draft microbial genomes using high-throughput sequencing data from infected host tissues, even without a host genome sequence. Our initial assembly, performed using the complete pooled sequencing dataset from both Trachelipus rathkei individuals, was initially highly fragmented; even the contigs derived from Rickettsiella were short and numerous. The mixture of sequencing reads from two relatively similar bacterial genomes, at different depths, likely resulted in a highly “tangled” de Bruijn graph. However, partitioning the initial assembly into different fractions representing the different bacterial taxa present, using both coverage and homology information, followed by iterated separate assemblies of the raw sequence reads that map to each fraction, resulted in substantially improved assemblies. For the Rickettsiella strain found at high depth in the female individual, the results were especially impressive: the N50 was greater than 384 kb, and the longest contig was over 550 kb (Table 1). There were large blocks of synteny between this assembly and the previously available R. grylli NZ_AAQJ00000000 assembly, and its total size was only slightly smaller than R. grylli (1.509 Mb vs. 1.581 Mb), supporting the quality of this assembly. Even though this approach relied on “seeding” the initial assembly with contigs showing similarity to the previously available R. grylli sequence, it was still able to recover novel genomic regions. It is hypothetically possible that these regions might be chimeric sequences, linking our Rickettsiella assembly to data from other bacteria that happened to be present in the DNA samples, as terrestrial isopods are known to harbor diverse microbiota (Dittmer et al., 2016). However, several lines of evidence argue against this: first, other regions flagged as assembly errors seem to be false positives, as they actually align nearly perfectly to the reference R. grylli genome (Figs. S1 and S3); second, sequencing depth in these candidate island regions was very high and similar to the rest of the R. isopodorum genome, and there are plenty of read pairs that map well to the borders of the candidate islands, with one member in the island and the other member outside it (Fig. S4); and third, sequence reads still align well to these regions flagged by REAPR, even though these areas identified as possible errors do have lower coverage (Fig. S3). Thus, any errors that are present in our R. isopodorum assembly are more likely to be either false positives or small-scale indels that would not alter our conclusions, rather than chimeric contigs leading to the incorrect predictions of genomic islands.

We also checked our assembly for the presence of potential contaminants, as metagenomic assembly is difficult, and it is plausible that sequences from other bacteria may have been incorporated into our assembly, perhaps explaining the presence of numerous short contigs. However, we found little evidence of this type of contamination; instead, the short, fragmented contigs appear to represent multi-copy sequences, which would be impossible to resolve without additional long-read sequence data. Even if some of these short contigs do represent contamination, their inclusion in the assembly would not alter any of our main conclusions below, as they are too short to contain any predicted horizontally transferred genes.

Phylogenetic diversity of Rickettsiella infecting terrestrial isopods

Our phylogenetic results provide clear support for two distinct lineages of Rickettsiella in terrestrial isopods: one representing R. isopodorum, described by Kleespies, Federici & Leclerque (2014); and one representing R. grylli, whose genome sequence is available in GenBank (NZ_AAQJ00000000) but for which little metadata or supporting information, such as host species and assembly methods, is available. While Dittmer et al. (2016) also found two distinct Rickettsiella lineages infecting the common pillbug Armadillidium vulgare, our multilocus sequencing data clearly link these two lineages to R. isopodorum and R. grylli NZ_AAQJ00000000. These two Rickettsiella lineages are probably distinct species, as the divergence between them is at least as great as the divergence between either one and Rickettsiella lineages found in other arthropod hosts, including insects and acari (Fig. 2). We would argue that the R. grylli strain infecting terrestrial isopods should probably be renamed, as it is likely distinct from R. grylli infections in crickets. While a given Rickettsiella lineage can probably infect multiple host species, genetic data show that Rickettsiella lineages infecting isopods are clearly distinct from those infecting insects (Kleespies, Federici & Leclerque, 2014; Fig. 2). Moreover, the genome of R. grylli in crickets was estimated to be 2.10 Mb in size (Frutos et al., 1989), which is quite distinct from the ∼1.5–1.6 Mb genome found in these isopod lineages. This unfortunate choice of names came about because R. grylli was first described in the cricket Gryllus bimaculatus (Vago & Martoja, 1963), and terrestrial isopod Rickettsiella infections came to be called R. grylli because of their phenotypic similarity to R. grylli, although they were initially proposed as R. armadillidii (Vago et al., 1970; Abd El-Aal & Holdich, 1987). Unfortunately, it is unclear which isopod-infecting Rickettsiella lineage these earlier papers deal with because of a lack of genetic data.

We speculate that these two lineages of Rickettsiella infecting terrestrial isopods may differ in pathogenicity. If this turns out to be true, we predict that the “R. grylli” may be a less pathogenic species than R. isopodorum, or perhaps even a neutral or beneficial endosymbiont. Unlike R. isopodorum, its DNA was found at a low density in our sequencing data, suggesting it did not proliferate as strongly within its host, although it could have just been in the early stages of infection. Although Rickettsiella is traditionally described as a potent pathogen (Dutky & Gooden, 1952; Hall & Badgley, 1957; Kleespies, Federici & Leclerque, 2014), recently, benign or even beneficial Rickettsiella endosymbionts have been found in other arthropod lineages (Tsuchida et al., 2010; Tsuchida et al., 2014; Iasur-Kruh et al., 2013), and some evidence suggests some forms of Rickettsiella may also be capable of vertical transmission (Iasur-Kruh et al., 2013). Consistent with this hypothesis, Dittmer et al. (2016) report that Rickettsiella was detected even in some isopods that did not show symptoms. Unfortunately, we do not have records of whether the individuals we sequenced showed any symptoms, but we do occasionally observe individuals filled with a milky white fluid (personal observations), a classic sign of Rickettsiella infection in isopods (Vago et al., 1970).

We do find some evidence of local geographic differentiation; isopod Rickettsiella isolates from upstate New York cluster into distinct sub-clades from those from other localities (Fig. 2). This is surprising because in a previous study, R. isopodorum isolates from two different host species in California and Germany were nearly indistinguishable (Kleespies, Federici & Leclerque, 2014). Indeed, these two samples from different continents cluster together to the exclusion of our R. isopodorum samples from upstate New York (Fig. 2). This discrepancy is unlikely to be explained by errors in our assembly, because independent samples from New York identified by PCR and Sanger sequencing, including some from different host species, also clustered with our assembled sequences, to the exclusion of the samples from California and Germany from Kleespies, Federici & Leclerque (2014) (Fig. 2B). Kleespies, Federici & Leclerque (2014) examined some live animals; it might be conceivable that some horizontal transmission of Rickettsiella occurred in the lab after specimens were caught from the wild, perhaps explaining why their samples from California are essentially indistinguishable from their German samples yet distinct from our New York samples. However, the branch lengths in these cases are much shorter than those separating different Rickettsiella species, and this pattern is based on only a small number of molecular markers, so further work is needed to confirm patterns of geographic differentiation.

It is clear, on the other hand, that horizontal transmission of Rickettsiella across different terrestrial isopod host species is probably common. For example, we obtained gidA sequences from “R. grylli” isolates from T. rathkei and Philoscia muscorum that were nearly indistinguishable; similarly, R. isopodorum gidA sequences from T. rathkei, Oniscus asellus, A. vulgare, and Porcellio scaber also formed one clear clade (Fig. 2).

Possible role of horizontal transfer and genomic islands in the evolution of pathogenicity

We used two approaches to identify candidate horizontal gene transfer events. HGTector (Zhu, Kosoy & Dittmar, 2014) identified several candidate horizontally transferred genes (Table 3). Most of these are enzymes whose biological roles are unclear. However, one of them, A1D18_00810, contained a predicted outer membrane autotransporter protein (Table 3); proteins in this family are often virulence factors (Henderson & Nataro, 2001).

Our custom comparative genomics approach to detect genomic islands, though relatively simple, also detected several interesting regions in the R. isopodorum draft genome. Although genomic islands are typically at least 10 kb in size (Langille, Hsiao & Brinkman, 2010), some of the candidate islands identified here are as small as ∼3.6 kb. As previously mentioned, these regions share similar sequencing depth to the rest of the R. isopodorum genome, and although REAPR found some possible assembly errors, several lines of evidence suggest that these are false positives or small-scale indels in the assembly rather than chimeric contigs (see above). In addition to displaying no homology to the R. grylli and D. massiliensis genomes, a few of these regions displayed GC content quite different from the rest of the assembly (Table 2), further supporting the hypothesis that these represent horizontally acquired genomic islands. Although some of these candidate islands might simply be regions that are undergoing rapid divergence, obscuring their homology to R. grylli sequences, most of them did contain predicted genes that had BLAST hits in more distantly related bacterial taxa, also suggesting that they are horizontally acquired. Moreover, most of these are unlikely to be the result of differential loss of ancestral genes in R. grylli and D. massiliensis, because they lacked any close BLAST hits in other members of the Legionellales as well.

The first two islands each contained several predicted genes with no apparent matches in genetic databases. It is possible that the gene predictions here might be erroneous, or that these candidate genomic islands may have been horizontally acquired from species that are currently not represented in the databases. Other candidate islands contained multiple predicted genes that had strong BLAST hits to predicted genes in other organisms, consistent with the idea that these regions represent horizontally acquired genetic material. For instance, one candidate island of about 7 kb contained multiple predicted enzyme genes matching a variety of other bacterial taxa belonging to clades with soil-dwelling members. Rickettsiella can be transmitted via soil even though it is an intracellular endosymbiont (Dutky & Gooden, 1952), so there are likely opportunities for it to exchange genetic material with other soil microbes.

Intriguingly, one R. isopodorum candidate island contains multiple predicted genes showing homology to genes from other taxa that have been implicated in toxicity. One of these genes, A1D18_02500, is partially outside the candidate island region and has a partial match to a D. massiliensis hypothetical protein, though no close matches in R. grylli, and it has better matches to proteins from other bacterial taxa; this gene contains a predicted anthrax toxin domain, and its matches are annotated as “adenylate cyclase” and “anthrax toxin”. A second predicted gene in this cluster contains a putative RING-finger domain, though this domain is associated with a variety of functions so its significance here is unclear. Finally, the third predicted gene in this cluster has clear homology to the Mcf2 gene, which is an insecticidal toxin found in Photorhabdus luminescens and Xenorhabdus nematophilus (Daborn et al., 2002; Waterfield et al., 2003); this gene has also been horizontally transferred into fungal symbionts of plants (Ambrose, Koppenhöfer & Belanger, 2014). Given the large phylogenetic distance between the R. isopodorum homolog of this gene and other known members in this gene family (Fig. 4), the donor may be an as-yet unsequenced, and perhaps uncultured, bacterium; alternatively, it may have been heavily modified by selection after its integration into R. isopodorum. The presence of Mcf-like proteins is sufficient to render E. coli toxic to insects (Daborn et al., 2002; Waterfield et al., 2003; Ambrose, Koppenhöfer & Belanger, 2014), making this an intriguing pathogenicity candidate gene in Rickettsiella isopodorum. The presence of this gene in R. isopodorum but not R. grylli is consistent with the admittedly speculative hypothesis that R. isopodorum may represent a more pathogenic lineage, but further work is needed to test this idea. For example, wild animals could be sampled, checked for symptoms, and checked for Rickettsiella infections using PCR and sequencing; an association between symptoms and R. isopodorum-like DNA, but not R. grylli-like DNA, would further support this hypothesis. Further whole-genome sequencing of independent Rickettsiella isolates from infected hosts using approaches similar to the one we adopted could shed more light on genomic variation in these bacteria and help identify candidate genes associated with variation in pathogenicity. Experimental infections using E. coli transformed with candidate pathogenicity genes would then shed light on these genes’ biological roles. Rickettsiella, and its terrestrial isopod hosts, might therefore make a good model system for studying the evolution of host-pathogen and host-symbiont interactions.

Supplemental Information

Supplemental Information 1 Supplementary figures & tables

Click here for additional data file.

Supplemental Information 2 Supplementary data—analysis code, processed data, etc

Click here for additional data file.

This work was completed in partial fulfillment of Y Wang’s undergraduate senior capstone project at SUNY Oswego. We thank K Alvarado, R Joachim, P Newell, A Walter, and the editor and reviewers for helpful suggestions on earlier versions of this manuscript. We also thank the National Center for Genome Analysis Support at Indiana University, the GCAT-SEEKquence Consortium, and V Buonaccorsi and C Walls at Juniata College for computing support.

Additional Information and Declarations

Competing Interests

Author Contributions

DNA Deposition

Data Availability

The authors declare there are no competing interests.

YaDong Wang conceived and designed the experiments, performed the experiments, analyzed the data, contributed reagents/materials/analysis tools, wrote the paper, reviewed drafts of the paper.

Christopher Chandler conceived and designed the experiments, performed the experiments, analyzed the data, contributed reagents/materials/analysis tools, wrote the paper, prepared figures and/or tables, reviewed drafts of the paper.

The following information was supplied regarding the deposition of DNA sequences:

The assembled genomic sequences are available at GenBank under accession numbers MCRF00000000 and LUKY00000000. The raw sequencing reads are available from the NCBI SRA under accession numbers SRR4000573 and SRR4000567.

The following information was supplied regarding data availability:

Raw data are available via NCBI SRA accession numbers SRR4000573 and SRR4000567.

Analysis code and processed data are available in Supplementary Materials.

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
