# Peer review of "Candidate pathogenicity islands in the genome of ‘Candidatus Rickettsiella isopodorum’, an intracellular bacterium infecting terrestrial isopod crustaceans"

_PeerJ, doi:10.7717/peerj.2806_

## Round 0.1 · original submission · Major Revisions

· Academic Editor

Major Revisions

Your manuscript was fully evaluated at the editorial level and by independent peer reviewers. The reviewers appreciated the attention to an important topic and the robustness of the genomic data but identified some aspects of the manuscript that should be improved.

Reviewer 2 has made several suggestions on how to strengthen the conclusions. I believe the evaluation of the presence of other bacterial species (which could confound the analyses), an attempt to improve the assembly, addressing the comments on genomic islands data analysis and clarified details of phylogenetic analyses should be incorporated in the revised manuscript.

Both reviewers have also minor comments, which I hope you will be able to address in your reply.

Reviewer 1 ·

Basic reporting

Basic form is solid, there were a few typos to be corrected:

Line 23 - arthropods is spelled athropods
Line 59 - Rickettsiella is spelled Rickettiella

Various places, there is no space between the number and unit; e.g. line 328 "10kb"

A general note regarding flow (take or leave): the paragraph that starts at line 290 seems it would fit better somewhere after line 338 where the rest of the discussion regarding pathogenicity is located.

Experimental design

Overall the design and analysis are excellent.

Validity of the findings

The overall assemblies, especially given the lack of reference genomes, is strong.

In order to better assess the genomic islands, in particular, the potential pathogenicity factors, it would be helpful to include a column with %ID/Positives.

The homology between the Rickettsiella protein and a cytotoxin is exciting, however, the statement that R. isopodorum may be neutral/beneficial is a little overstated for the evidence at this point (line 292). It is difficult to assess levels of pathogenicity without doing a broader study of numbers of infected individuals/mortality rates.

There was a difference between males and females, some discussion of why there might be a split between sexes and what that might mean for fecundity would be interesting.

·

Basic reporting

This manuscript by Wang and Chandler presents comparative genomic analyses between two species of Rickettsiella (Legionellales) infecting arthropods – R. grylli and R. isopodorum. In terms of genomic analyses, the Rickettsiella genus has been neglected for a long time, so such study is much needed and I have no doubt it will be of broad interest. The results show that (pan)genomes of Rickettsiella species are very dynamic similarly to genomes of other pathogens or facultative symbionts of arthropods and seem to be influenced by horizontal gene transfer with unrelated arthropod-infecting or soil bacteria. I applaud the authors for sharing all of their Bash, R, and Python scripts (although hard-coded for the Rickettsiella data) in supplementary data and PeerJ for requiring this.

Unfortunately, an isopod species (Trachelipus rathkei) with two distinct lineages of Rickettsiella was selected for genome sequencing. This has caused complications in the genome assembly, but I consider the methodology and interpretation of the data to be solid given the data. However, although it is stated at the end of introduction that the aim of the study was to explore genomic variation among Rickettsiella isolates in terrestrial isopods, the study just scratches the surface and most features of the genomes were not analyzed at all, so I have several suggestions for improvements of the experimental design.

Experimental design

Major comments
(1) Presence of other bacterial taxa
The total assemblies were not evaluated for presence of other bacterial species. I would strongly recommend to run this kind of analysis [I greatly recommend Blobtools; https://blobtools.readme.io/] to be certain that some of the short contigs included in the separated Rickettsiellla assemblies do not come from other sources. Taxon-annotated GC-coverage plots generated by Blobtools can be also used to detect potential horizontal gene transfers of Rickettsiella genes into the isopod genome from the true infection.

(2) Genome assembly and analyses
The best assembly is still very fragmented (198 contigs) and contains numerous short contigs. Apart from getting better data such as long reads from PacBio/Oxford Nanopore, there are several other ways how to get the assembly into a better shape, for example using a metagenome assembler with merged long k-mers usually helps in my experience [e.g. SPAdes meta http://spades.bioinf.spbau.ru/release3.9.0/manual.html#meta or computationally less demanding Megahit https://github.com/voutcn/megahit]. Additionally, since average coverage in the female sample was ~1000x, random down-sampling of the data to <100x can remove majority of reads from the other species and authors can completely avoid their mapping and assembly strategy.

The article does not provide any details about possible causes of the fragmented state of the assembly other than the presence of the second species. Were there any plasmids or phages (or their marker genes) detected in the assembly? PlasmidSPAdes [http://spades.bioinf.spbau.ru/release3.9.0/manual.html#plasmid] allows to assemble such high copy number extrachromosomal sequences from raw sequencing data only. In a comparative genomic analysis such as this, I would also appreciate to see an analysis of insertion sequences, prophage sequences (e.g. ISfinder/ISsaga [http://issaga.biotoul.fr/] and PHASTER [http://phaster.ca/] work with draft assemblies), and any other duplicated regions such as rRNA operons breaking up the assembly. Basic genome statistics such as total number of tRNAs, ncRNAs, protein-coding genes, etc. should be also included in Table 1. I understand that pseudogene calls are not very reliable in the draft assembly, but at least a preliminary analysis of Rickettsiella metabolic pathways would be interesting for readers working with other arthropod-associated bacteria such as Wolbachia.

(3) Analyses of genome islands
The presence of the putative horizontally acquired makes caterpillars floppy (mcf) gene in the Rickettsiella genome is perhaps the most exciting finding from the genome island analysis, but it also deserves more evidence. Please infer a phylogenetic tree for this gene (including the homologue found in fungal symbionts of Epichloe grasses) to infer potential donors of this HGT. Additionally, I see no particular reason to focus on HGTs in only the genomic islands as blast-based methods such as HGTector [https://github.com/DittmarLab/HGTector] can quickly evaluate HGT candidates for all proteins in a proteome and this genome-wide analysis would strengthen up the manuscript.

Minor comments

1) Why was Tamura-Nei model used in ML phylogenetic analyses? Was any model-selection statistical method used to prefer this model over other models?

2) Why was only D. massiliensis used as an outgroup and not other Legionellales? Having richer taxon sampling of other Legionellales may be essential to detect differential gene loss in the two Ricketsiella species as opposed to gene acquisition of genomic islands.

Validity of the findings

I consider the genome data presented in the paper robust, but the main concern I have is that the genome was not analyzed in much detail as the paper focuses mostly on its assembly.

Additional comments

Line 40: use 16S rRNA gene instead of 16s rRNA gene
Line 241: Assembly of bacterial genomes from mixed species data
This paragraph seems too long as almost identical strategies were already described by numerous previous studies.
Line 424: Space-efficient
Line 434: missing issue, pages, and DOI
References: several species names such as Armadillidium vulgare are not in intalics.

---

## Round 0.2 · Minor Revisions

· Academic Editor

Minor Revisions

Thank you very much for the thoughtful revision. All the previous concerns have been adequately addressed, the manuscript presents a solid piece of research and as soon as the minor comments of the Reviewer 2 on the new version are addressed I will be delighted to accept it for publication.

·

Basic reporting

The authors have adequately addressed all my comments and I have no further major issues with the MS. The only minor comment I have is about the newly inferred tree and its evolutionary interpretations (Figure 4).

It should be either stated in the figure legend that the tree is unrooted or I would suggest to show the tree in circular net-like format to avoid any misunderstandings as remaining trees shown in the manuscript (Figure 2) appear to be rooted (by Diplorickettsia outgroup). Moreover, the inferred tree and its underlying sequence alignment strongly suggest that the mcf-like sequence from Rickettsiella is very distant and only partially homologous to other mcf sequences in genbank. Interpretations of its evolutionary history were appropriately toned down in the discussion and results sections by the authors, but I would also suggest to mention the poor and partial homology in the abstract.

Experimental design

No Comments

Validity of the findings

No Comments

---

## Round 0.3 · accepted · Accept

· Academic Editor

Accept

Thank you very much for the review. I believe this paper is a solid piece of research, interesting from the perspective of evolution of symbiont genomes. Congratulations!